# Re-Whole Brain Radiotherapy May Be One of the Treatment Choices for Symptomatic Brain Metastases Patients

**DOI:** 10.3390/cancers14215293

**Published:** 2022-10-27

**Authors:** Takashi Ono, Kenji Nemoto

**Affiliations:** Department of Radiation Oncology, Faculty of Medicine, Yamagata University, 2-2-2, Iida-Nishi, Yamagata 990-9585, Japan; knemoto@ca3.so-net.ne.jp

**Keywords:** re-whole brain radiotherapy, biological effective dose, late toxicities

## Abstract

**Simple Summary:**

Some institutions consider re- whole brain radiotherapy (WBRT) administration for brain metastases recurrence; however, there is insufficient information regarding this. We aimed to review re-WBRT administration among these patients. symptomatic improvement was sometimes observed, with tolerable acute toxicities. Therefore, re-WBRT may be a treatment option for patients with symptomatic recurrence of brain metastases. However, asymptomatic patients should not undergo re-WBRT owing to the unclear toxicity data. Although there is no strong evidence, 20 Gy in 10 fractions or 18 Gy in five fractions may be a reasonable treatment method within the tolerable total biological effective dose 2 ≤ 150 Gy.

**Abstract:**

Generally, patients with multiple brain metastases receive whole brain radiotherapy (WBRT). Although, more than 60% of patients show complete or partial responses, many experience recurrence. Therefore, some institutions consider re-WBRT administration; however, there is insufficient information regarding this. Therefore, we aimed to review re-WBRT administration among these patients. Although most patients did not live longer than 12 months, symptomatic improvement was sometimes observed, with tolerable acute toxicities. Therefore, re-WBRT may be a treatment option for patients with symptomatic recurrence of brain metastases. However, physicians should consider this treatment cautiously because there is insufficient data on late toxicity, including radiation necrosis, owing to poor prognosis. A better prognostic factor for survival following radiotherapy administration may be the time interval of >9 months between the first WBRT and re-WBRT, but there is no evidence supporting that higher doses lead to prolonged survival, symptom improvement, and tumor control. Therefore, 20 Gy in 10 fractions or 18 Gy in five fractions may be a reasonable treatment method within the tolerable total biological effective dose 2 ≤ 150 Gy, considering the biologically effective dose for tumors and normal tissues.

## 1. Introduction

The incidence rate of brain metastasis (BM) was 2.0% in all patients with cancer and 12.1% among patients with metastatic disease during 2010–2013, according to the Surveillance, Epidemiology, and End Results data [1]. Moreover, 1.7% of patients had synchronous BM at the first diagnosis [2]. The lung, breast, and skin melanoma are common sources [3] of BM among adult patients; the occurrence rate of BM is the highest (13.5%) among lung cancer patients [4]. The prognosis of patients with BM is poor, with a median survival time (MST) of 3.0–5.0 months [1,5].

The first-line treatment for patients with single BM with favorable survival is surgery following whole brain radiotherapy (WBRT) [6]. However, WBRT is sometimes not considered because of its neurocognitive toxicity [7]. Stereotactic radiosurgery (SRS) is recommended as an alternative to surgical resection, especially when surgical resection is likely to induce new neurological deficits [8]. The routine use of chemotherapy is not recommended for patients with multiple BM who are unsuitable for local therapy [9]. Routine use of immunotherapy and molecular targeting therapy is also not recommended for these patients due to insufficient evidence [10], although some reports suggest their efficacy [11]. WBRT is recommended for multiple BM [7].

The standard treatment dose scheme of WBRT is 30 Gy in 10 fractions and 20 Gy in five fractions; however, there is no evidence that higher doses improve survival or BM control [12]. More than 60% of patients received complete or partial responses, and >50% of patients showed symptom relief. However, 35–52% of patients showed deterioration in cognitive tests 3–6 months after WBRT [13]. For increasing local control, some trials suggested that WBRT with concomitant temzolomide improved local control of BM without survival benefit [14,15]. Some trials have attempted to reduce cognitive function using memantine and hippocampus sparing intensity-modulated radiotherapy [16,17,18].

Although the survival of patients with multiple BM is poor, some patients experience recurrence with neurocognitive symptoms. If the incidence of recurrent BM is reduced, reirradiation using SRS can be adaptive, and some reports have suggested its efficacy and tolerability [19,20]. However, none of the institutions perform salvage SRS because it requires a precise setup [21]. Therefore, re-WBRT has become a treatment option for recurrent BM, especially for multiple recurrences of BM, and has been administered since the 1970s [22].

No review has focused only on re-WBRT; therefore, in this narrative review, we aimed to focus on re-WBRT, including recent findings.

## 2. Treatment Efficacy

The characteristics of previous reports are shown in Table 1 [22,23,24,25,26,27,28,29,30,31,32,33,34,35,36,37,38,39,40]. All reports were retrospective in nature, the median age of the patients was 48–60 years, the median time between administration of the first WBRT and re-WBRT was 5–29 months, and the majority of cases were of lung cancer (including non-small cell lung cancer [NSCLC] and small cell lung cancer [SCLC]), breast cancer, and melanoma.

There was 24–74% symptom improvement (Table 2); these data are highly variable. However, the deterioration rate was less variable at 0–38%. If patients with symptomatic BM do not receive any treatment, their symptoms and performance statuses worsen; they do not improve or stabilize. Progression of BM worsens the quality of life (QOL), although there are no reports on the quantitative evaluation of QOL before and after re-WBRT; more than 50% of patients maintain stable disease. However, causes of data variation include heterogeneous data and differences in the criteria for determining the response to re-WBRT. 

Some articles reported that 24–32% of patients experienced complete resolution of neurocognitive function after re-WBRT [23,26,31,34]. However, Hazuka et al. reported no complete resolution with short survival [24]. Therefore, they did not recommend re-WBRT, although it was recommended in most articles. Although many patients did not respond to re-WBRT, Hazuka et al. reported that some patients showed symptom improvement (only one patient survived more than 1 year with the median survival time of 8 weeks). Furthermore, Suzuki et al. reported that re-WBRT or gamma knife treatment significantly improved BM control [39]. Therefore, re-WBRT may be a treatment option for patients with neurocognitive disturbances due to BM recurrence.

Further, the effect of re-WBRT does not persist for a long time. Kurup et al. reported the details of the response duration [23]. In this report, 62% of responders survived less than 3 months, 29% survived 3–6 months, and 9% survived 6–12 months after re-WBRT, and the responders did not maintain their improved status over 12 months after re-WBRT. This duration was inferior to the first WBRT, wherein 6% of patients maintained their improved status for more than 12 months. The median response duration was 2.5–2.75 months [23,26].

Although steroid use may influence these data, Suzuki et al. reported that patients receiving radiotherapy tended to achieve better local control than those receiving the best supportive care or chemotherapy (hazard ratio was 0.533, *p* = 0.050) [39]. Therefore, re-WBRT may improve neurocognitive symptoms by increasing local control.

In summary, regarding efficacy, re-WBRT may be a treatment option for improving or maintaining the QOL of patients with BM. However, physicians should be cautious about its effects and the short response duration.

## 3. Overall Survival and Prognostic Factors

The MST in previous reports was 2.0–6.9 months (Table 2). Although some variations were observed, the prognosis was poor.

The longest survival time after re-WBRT was 12 to 72 months [23,26,28,30,32,33,37]. Li et al. reported the case of a patient with lung adenocarcinoma approximately 7 years after re-WBRT [41]; the first WBRT was administered at 40 Gy in 20 fractions along with a boost with 10 Gy in five fractions, and re-WBRT was administered at 30 Gy in 15 fractions plus boost intensity-modulated radiation therapy with 20 Gy in 10 fractions after the first WBRT. This patient lived for approximately 12 years after the first WBRT and had a Karnofsky performance status (KPS) of 80 without deterioration in QOL; however, there was a mild neurocognitive deficit. Some patients lived longer, although most lived only for a short time.

First, there are ambiguous findings on whether adding re-WBRT improves survival. Suzuki et al. reported that patients receiving radiotherapy, including re-WBRT and gamma knife therapy had a better prognosis compared to patients receiving the best supportive care or chemotherapy only [39]. Although this study included only a small number of patients with SCLC, patients with recurrent BM of SCLC may benefit from re-WBRT. Moreover, re-WBRT may maintain patients’ QOL, as mentioned above.

Previous reports suggested various negative prognostic factors using multivariate analysis. The prognostic factors suggested in two or more reports were the primary sites of SCLC [33,37], presence of extracranial metastasis [30,37,39], low KPS (<60–80) [30,32,35,37,38,40], and a short time between the first WBRT and re-WBRT [32,37].

Logie et al. reported the largest amount of re-WBRT and described the reirradiation score [37]. In this score, the following five factors were used: primary site of SCLC, presence of extracranial metastasis, KPS < 80, the interval between two radiation therapy courses < 9 months, and uncontrolled primary. These five factors had one point each, and the score was divided into three groups. The MST of these groups was significantly different, and scores 1–2, 3, and 4–5 denoted 7.2, 3.0, and 2.2 months, respectively. There were no patients with a score of 0 in this study, and most patients who received re-WBRT had negative prognostic factors in real clinical situations. Regarding this score, Burr et al. reported that the score of 1–2 was significantly associated with improvement in overall survival compared to the score of 3–4 (MST was 4.6 months vs. 2.9 months, respectively) [40]. Although these reports were also retrospective in nature, the reirradiation score was a helpful criterion for clinicians. However, this score did not predict very short survival durations, i.e., ≤1 month. For these patients, the palliative prognostic score or palliative prognostic index, which is used in palliative care to predict survival of 21–30 days, may be suitable [42,43].

Ozgen et al. reported that the interval of less than 9.5 months between two radiotherapy courses was a negative factor for survival [32]. In contrast, Aktan et al. and Burr et al. reported that the interval of less than 9 months between two radiation therapy courses was not a significant factor for survival [35,40]. In addition, Minninti et al. reported that the interval of less than 12 months between two radiation therapy courses was not a significant factor for survival [39]. However, these contrasting findings were based on a smaller number of patients compared with those examined by Logie et al. [37]. Although physicians should be cautious about the use of these data, 9 months may be one of the indications for survival.

Akiba et al. and Ozgen et al. reported that patients with lung cancer lived longer than those with breast cancer [30,32]. However, the number of patients with breast cancer was small compared to those with lung cancer. In another study, Scharp et al. and Logie et al. reported that SCLC was a poor prognostic factor for survival [33,37]. Some reports did not show significant differences in the primary site for survival [26,27,28,32,34,35]. Moreover, most reports on re-WBRT included patients who were not affected by lung cancer, breast cancer, or melanoma. Therefore, SCLC may be one of the worst prognostic factors for survival after re-WBRT, although this finding requires further research.

Data on histopathology were only reported in the article. Akiba et al. performed a statistical analysis on the differences in survival between patients with adenocarcinoma, small cell carcinoma, and large cell carcinoma [30]. However, the difference was not statistically significant.

In summary, most patients did not live longer than 12 months after re-WBRT, although a limited number of patients lived longer. Considering radiotherapy, the interval of >9 months between the first WBRT and re-WBRT may be a better prognostic factor. The re-irradiation score can be an indicator of prognosis after re-WBRT.

## 4. Toxicities

In previous reports, there were few severe toxicities reported, except for 4% of grade 3 leukoencephalopathy cases reported by Minniti et al. [34]. As shown in Table 3, the most frequent acute toxicities were fatigue and headache. However, toxicities, including grade and late toxicities, have not been well documented. This may be because most of the patients were not followed up due to poor conditions.

Regarding acute toxicities, Shehata et al. reported that 14% of patients had severe nausea, vomiting, and headaches, although the grade was not determined because the details of their symptoms and treatment were not shown [22]. However, all patients recovered within 48 h. Acute reactions are difficult to evaluate because the disease causes nausea, vomiting, and headaches [23]. Therefore, acute toxicities were tolerable, as most reports stated, and the symptoms could be treated, although it is sometimes challenging to diagnose toxicities or disease symptoms.

Central nervous system necrosis due to radiation is an important concern for physicians when considering re-WBRT. These data were also not well documented in most reports and were only reported in two articles. Kurup et al. reported that one patient experienced seizures and visual problems 7 months after re-WBRT because of radiation necrosis detected by imaging [23]. In contrast, Hazuka et al. reported that there were three patients with radiation necrosis based on data from eight necropsy cases [24]. Two of the three patients deteriorated neurologically prior to death. These two patients lived for 11 and 20 months, respectively. There was no data on these patients’ severity and treatment details; thus, their grades were not estimated. Therefore, patients living > 6 months may experience symptoms of radiation necrosis due to re-WBRT.

Re-WBRT administration for patients with unsympathetic multiple BM is difficult because these patients may have only morbidity due to re-WBRT. BM can be monitored using systemic therapy among these patients; it may be the first therapeutic option for BM from highly chemotherapy sensitive primary tumors, such as germ cell tumors and SCLC [44]. Suzuki et al. reported that ≥ 4 cycles of chemotherapy before the recurrence of BM and chemotherapy after the recurrence of BM were significantly better prognostic factors for the survival of patients with SCLC [39]. They also reported that chemotherapy after BM recurrence was a significantly better prognostic factor for BM control after brain failure. Moreover, there is increasing evidence that molecular targeted therapies or immunotherapy can treat BM [44]. Multiple clinical trials demonstrated that the central nervous system objective response rates are > 50% when using newer generation tyrosine kinase inhibitors, and treatment strategies involving upfront central nervous system active tyrosine kinase inhibitors alone with close magnetic resonance imaging surveillance may be considered for some patients with asymptomatic BM [45]. Therefore, unsympathetic patients who are adaptable to systemic therapy may be suitable for systemic therapy before re-WBRT is considered.

In summary, physicians should be aware that previous reports may have underestimated toxicities, especially late toxicities, because of poor prognosis after re-WBRT or publication bias, although many previous reports showed that most toxicities after re-WBRT were tolerable, as mentioned above. Additionally, physicians should not consider re-WBRT among patients without symptoms because toxicities, including radiation brain necrosis, are only observed after treatment. For these patients, upfront systemic therapy may become the preferred treatment if they are suitable for it.

## 5. Dose Prescription and Total Dose

The median dose of re-WBRT was variable, as shown in Table 4, although most patients received the first WBRT at 30 Gy in 10 fractions, as previously reported. The median total dose of re-WBRT ranged from 20 to 30 Gy, and a lower dose was used in most reports compared to the first WBRT. When the sum of the dose of first WBRT and re-WBRT was calculated, the biological effective dose (BED) was useful. BED was calculated using the linear quadratic model and was indicated as follows: nd (1 + d/[α/β]) Gy (d = dose per fraction, n = number of treatments) [46]. Table 5 shows the sum of the initial WBRT and re-WBRT doses (α/β = 2 for brain [BED 2] and α/β = 10 for tumor control [BED 10]).

Whether a higher BED improves or worsens overall survival and BM control rate requires further research. Previous studies investigated the correlation between survival and total BED, but there are no reports suggesting that a higher total BED improves survival after re-WBRT [32,35]. Ozgen et al. also reported that total BED administration did not improve the symptomatic response rate [32]. Although there are no data on the correlation between total BED and BM control, a higher BED will not improve BM control considering survival and symptom data. Even for the first WBRT, there is no relationship between survival and BED [12]. Further, no studies have investigated the correlation between a higher BED and prognosis.

If a higher BED does not improve the prognosis, the tolerance dose should be considered during re-WBRT administration. Regarding the median total BED, Akiba et al. reported the highest median BED 2 as 150 Gy [30]. They reported details of toxicity, including brain atrophy and cognitive function, and there were no grade ≥ 3 toxicities after re-WBRT. Hazuka et al. reported sympathetic radiation central nervous necrosis, as mentioned above [24]. The total BED 2 was 171 Gy (96 Gy + 75 Gy) and 162.45 Gy (75 Gy + 87.45 Gy), respectively. Patients who lived for 7 years with mild neurocognitive deficits after re-WBRT received a total BED 2 = 140 Gy for the whole brain [41]. Although this patient received a booster dose for a small region, the expected maximum total BED 2 was 200 Gy, which was lower than the reirradiation tolerable dose of partial brain irradiation for glioma [47]. Considering that other reports also showed tolerable toxicity data, total BED 2 = 150 Gy may indicate a tolerable dose for re-WBRT.

The effectiveness of low-dose BED for recurrent BM should be considered for re-WBRT if a higher BED does not improve efficacy. In previous reports, the smallest median BED 10 of re-WBRT was 24 Gy (20 Gy in 10 fractions), the improvement ratio of neurocognitive symptoms was 33–60% and of MST was 2.8–4.0 months [26,33,36,38,39]. These data were not inferior to the previous data employing a higher BED 10 of re-WBRT. Moreover, a low dose per fraction leads to a low BED of 2 and better tolerability for re-WBRT. Therefore, 20 Gy in 10 fractions (BED 2 = 40 Gy) may be a treatment option for re-WBRT. However, the treatment period is of 2 weeks. Although there are no reports, 18 Gy in five fractions (BED 2 = 50.4 Gy, and BED 10 = 24.4) may be an alternative method for patients with poorer conditions, considering equal tumor control and brain tolerability.

There are no data supporting that higher doses improve patients’ prognosis or symptoms; thus, lower doses, as previously described, may be better for re-WBRT, considering the efficacy and toxicities.

Regarding other approaches, there are two unique re-WBRT methods. Abdel-Wahab et al. reported that 30 Gy in 20 fractions at 1.5 Gy per fraction was administered twice daily with a 6-hour interval [27]. In contrast, Burr et al. reported reduced dose rate methods using 0.067 Gy/min [40]. Both methods expected a reduction in radiation necrosis while maintaining tumor control. Although both reports suggested tolerable treatments, the potential benefits of these methods are unknown. Further, the treatment duration of one day is longer than the normal treatment duration, and the treatment burden for patients increases.

In summary, there is no clear evidence to improve clinical results by increasing the prescription dose of re-WBRT. Considering previous reports, the tolerable total BED 2 may be ≤ 150 Gy. Regarding the concrete dose scheme of re-WBRT, 20 Gy in 10 fractions or 18 Gy in five fractions may be a reasonable treatment dose, considering the toxicity and effects of BED.

## 6. Steroid Administration

Steroids are often used when patients receive re-WBRT for recurrent BM [24,27,28,30,31,36,38,40]. Except for three articles, most did not reveal the details of steroid prescriptions. Son et al. reported that the daily dose of steroids ranges from 4 to 24 mg [31]. Aktan et al. also reported that the daily dose ranged from 4 mg to 24 mg, with a median dose of 8 mg. Burr et al. reported a median dexamethasone dose of 4 mg, and patients who received > 4 mg of dexamethasone had a worse prognosis [40]. This may be because patients with severe symptoms received a higher dose of dexamethasone. They also reported that dexamethasone use decreased in 18% of patients, remained the same in 54%, and increased by 28% after re-WBRT. Therefore, they concluded that dexamethasone was a surrogate for clinical symptoms. 

In practical guidelines, a starting dose of 4–8 mg (a higher dose of 16 mg or more is recommended for patients with severe symptoms) per day of dexamethasone is recommended for patients who are sympathetic to BM [48]. Although it is unclear whether steroids improve the prognosis of patients undergoing re-WBRT for recurrent BM, dexamethasone administration should be considered because it can improve symptoms and toxicities from radiotherapy [49]. However, the side effects of dexamethasone increase with time. Therefore, determining a generalized dose scheme is difficult [49].

## 7. Concurrent Chemotherapy

There are no comparative studies examining the concurrent combination of chemotherapy on improved prognosis among patients who undergo re-WBRT. Minniti et al. reported the administration of re-WBRT and concurrent temozolomide, which has been approved for use in Italy [34]. In this article, the 12-month overall survival rate was 20%, which was not significantly higher than that in previous reports. Moreover, there was one patient with grade 3 leukoencephalopathy (the details of this case are not shown in this article). Therefore, there was no evidence of concurrent combination chemotherapy.

## 8. Meningeal Dissemination

Some reports included patients with meningeal dissemination [36,40], but there was only one report by Guo et al. focusing on this [36]. They reported that six patients showed meningeal dissemination (half of them had it during the first WBRT, and the others had it during re-WBRT). The MST for them was 1.5 months compared to 3.1 months for patients without meningeal dissemination. However, there was insufficient information regarding the ratio of improvement and toxicity.

Shafie et al. reported that 75.5% of patients showed improvement in neurological symptoms [50]. Although Morris et al. reported that WBRT did not prolong survival [51], Zhen et al. suggested that WBRT was an independent prognostic factor for patients with epidermal growth factor receptor type [52]. Re-WBRT may help in symptom improvement, although there is insufficient information regarding this.

## 9. Conclusions

Re-WBRT may improve symptoms due to BM recurrence with tolerable toxicities in patients with short-term survival. However, asymptomatic patients should not undergo re-WBRT owing to the unclear toxicity data. Although there is no strong evidence, 20 Gy in 10 fractions or 18 Gy in five fractions may be a reasonable treatment method within the tolerable total BED 2 ≤ 150 Gy. 

There is an ongoing clinical trial named evaluation of repeated whole brain radiotherapy versus best supportive care for multiple brain metastases (ERASER) (NCT03288272), and this data was hoped.

## Figures and Tables

**Table 1 cancers-14-05293-t001:** The characteristics of re-whole brain radiotherapy.

	Number of Patients	Median Age (Range) (Years)	Median KPS (Range) (%)	Median Time from First WBRT (Range) (Months)	Primary Site
Shehata et al. [22] ^a^	35	-	-	-	lung (49%), breast (19%), others (32%)
Kurup et al. [23]	56	53 (22–85)	-	5 (1–46)	lung (45%), breast (27%), others (28%)
Hazuka et al. [24]	44	54 (27–83)	-	7.8 (2–40)	NSCLC (34%), SCLC (20%), melanoma (11%), breast (9%), others (26%)
Cooper et al. [25]	52	Mean 57.3 (29–84)	-	-	lung (58%), breast (13%), melanoma (12%), others (17%)
Wong et al. [26]	86	58 (31–81)	-	7.6 (1.5–50.6)	lung (36%), breast (36%), colon (7%), melanoma (6%), others (15%)
Abdel-Wahab et al. [27]	15	Mean 51 (35 –73)	-	No data (1.9–28)	lung (16%), others (84%)
Sadikov et al. [28]	72	56.5 (34–75)	80	9.6 (2–37.3)	NSCLC (56%), breast (24%), SCLC (10%), others (10%)
Karam et al. [29]	37 ^b^	48	60 (40–90)	29 (4–58)	breast (100%)
Akiba et al. [30]	31	56 (38–74)	80	10 (2–69)	lung (84%), breast (16%)
Son et al. [31]	17	Mean 59	-	17.6 (3.6–46.9)	NSCLC (35%), SCLC (35%), breast (24%), colon (6%)
Ozgen et al. [32]	28	52 (36–68)	60 (50–100)	9.5 (3–27)	lung (61%), breast (39%)
Scharp et al. [33]	134	57 (31–82)	70 (40–100)	13.4 (3.4–58.5)	lung (87%), breast (9%), others (4%)
Minniti et al. [34]	27	54 (39–70)	70 (60–100)	15 (6.5–37)	NSCLC (67%), breast (33%)
Aktan et al. [35]	34	60 (32–76)	80 (50–100)	12.8 (5.8–45.7)	lung (65%), breast (21%), others (14%)
Guo et al. [36]	49	55 (29–77)	70 (40–90)	11.5 (1.5–49.2)	NSCLC (39%), SCLC (24%), breast (18%), melanoma (6%), others (12%)
Logie et al. [37]	205	55 (25–83)	60	9.1 (0.5–68.3)	NSCLC (41%), breast (31%), SCLC (16%), others (12%)
Bernhardt et al. [38]	67	-	Mean 60 (30–90)	14 (4–42)	SCLC (100%)
Suzuki et al. [39]	14	-	≥70	-	SCLC (100%)
Burr et al. [40]	75	54 (26–72)	80	9.7	breast (36%), NSCLC (25%), SCLC (12%), others (27%)

Abbreviations: KPS, Karnofsky performance status; WBRT, whole brain radiotherapy; NSCLC, non-small cell lung cancer; SCLC, small cell lung cancer. a: Most of data in this article included patients who received first WBRT and second or more WBRT. b: Including WBRT, partial brain radiotherapy, and stereotactic radiosurgery.

**Table 2 cancers-14-05293-t002:** The efficacy of re-whole brain radiotherapy.

	The Response for Neurocognitive Symptom				Steroid Use	MST (Months)	6 Months OS
	Improve	Stable	Deteriorated	Not Evaluable			
Shehata et al. [22]	69%	31%	0	0	-	-	-
Kurup et al. [23]	74%	13%	13%	0	-	3.5	-
Hazuka et al. [24]	27%	41%	14%	18%	100%	2	-
Cooper et al. [25]	42%	52%	6%	0	-	4.5	-
Wong et al. [26]	60%	29%	10%	1%	-	-	-
Abdel-Wahab et al. [27]	60%	33%	7%	0	Most patients	3.2	-
Sadikov et al. [28]	31%	21%	25%	23%	97%	4.1	-
Karam et al. [29]	14%	41%	19%	26%	-	6.9	-
Akiba et al. [30]	68%	-	-	-	84%	4	-
Son et al. [31]	47%	12%	0	41%	29%	5.2	-
Ozgen et al. [32]	39%	-	-	-	-	3	-
Scharp et al. [33]	39%	44%	17%	0	-	2.8	-
Minniti et al. [34]	63%	22%	15%	0	-	6.2	53%
Aktan et al. [35]	24%	38%	38%	0	59%	5.3	-
Guo et al. [36]	27%	24%	29%	20%	-	3	-
Logie et al. [37]	-	-	-	-	-	3.6	-
Bernhardt et al. [38]	40%	-	-	4%	72%	3	-
Suzuki et al. [39]	-	-	-	-	-	-	21%
Burr et al. [40]	18%	54%	28%	-	65%	4.1	-

Abbreviations: MST, median survival time; OS, overall survival.

**Table 3 cancers-14-05293-t003:** The toxicities after re-whole brain radiotherapy.

	>Grade 2	Headache	Nausea	Fatigue	Radiation Necrosis	Leukoencephalopathy	Brain Atrophy	Cognitive Disturbance
Shehata et al. [22]	-	14% ^b^	14% ^b^	-	-	-	-	-
Kurup et al. [23]	-	-	18%	-	2%	-	-	-
Hazuka et al. [24]	-	-	-	-	7%	-	-	-
Cooper et al. [25]	0	-	-	-	-	-	-	-
Wong et al. [26]	0	-	-	-	-	1%	1%	1%
Akiba et al. [30]	0	29%	26%	-	-	32% ^b^	28%	32% ^b^
Son et al. [31]	0	24%	24%	35%	-	-	-	-
Ozgen et al. [32]	0	-	-	-	-	-	-	-
Scharp et al. [33]	-	8%	7%	13%	-	-	-	10%
Minniti et al. [34]	4% ^a^	11%	7%	70%	-	26%	-	-
Bernhardt et al. [38]	0	34%	-	90%	-	-	-	-
Burr et al. [40]	0	17%	-	23%	-	-	-	3%

a: Grade 3 leukoencephalopathy. b: Occurrence rate was included 2 categories in 2 articles.

**Table 4 cancers-14-05293-t004:** Dose parameter of previous reports.

	Median Dose of First WBRT (Range)	Median Dose of Re-WBRT (Range)
Shehata et al. [22]	10 Gy	-
Kurup et al. [23]	-	20 Gy
Hazuka et al. [24]	30 Gy (30–36 Gy)	25 Gy (6–36 Gy) ^a^
Cooper et al. [25]	30 Gy	25 Gy
Wong et al. [26]	30 Gy (20–50.4 Gy)	20 Gy (7.9–30.6 Gy)
Abdel-Wahab et al. [27]	30 Gy (30–55Gy)	30 Gy ^b^
Sadikov et al. [28]	20 Gy (20–30 Gy)	25 Gy (15–25 Gy)
Karam et al. [29]	20 Gy	-
Akiba et al. [30]	30 Gy (26–42 Gy)	30 Gy (3–40 Gy)
Son et al. [31]	35 Gy (28–40 Gy)	21.6 Gy (14–30 Gy)
Ozgen et al. [32]	30 Gy (20–30 Gy)	25 Gy (20–30 Gy)
Scharp et al. [33]	30 Gy (30–40 Gy)	20 Gy (2–30 Gy)
Minniti et al. [34]	30 Gy	25 Gy
Aktan et al. [35]	30 Gy (25–30 Gy)	25 Gy (20–30 Gy)
Guo et al. [36]	30 Gy (20–37.5 Gy)	20 Gy (14–30 Gy)
Logie et al. [37]	20 Gy (12–48 Gy)	20 Gy (4–30.6 Gy)
Bernhardt et al. [38]	30 Gy (30–31.6 Gy)	20 Gy (20–30 Gy)
Suzuki et al. [39]	30 Gy (20–36 Gy)	20 Gy
Burr et al. [40]	30 Gy (24–51.35 Gy)	26 Gy (24–30 Gy)

Abbreviations: WBRT, whole brain radiotherapy. a: Ratio of patients who received WBRT was 84%. b: 30 Gy in 20 fractions (twice daily).

**Table 5 cancers-14-05293-t005:** Cumulative biological effective dose of previous reports.

	BED 2 (Initial + Second)	BED 10 (Initial + Second)
Cooper et al. [25]	131.25 Gy (75 Gy + 56.25 Gy)	70.25 Gy (39 Gy + 31.25 Gy)
Wong et al. [26]	115 Gy (75 Gy + 40 Gy)	63 Gy (39 Gy + 24 Gy)
Abdel-Wahab et al. [27]	127.5 Gy (75 Gy + 52.5 Gy)	73.5 Gy (39 Gy + 34.5 Gy)
Sadikov et al. [28]	116.25 Gy (60 Gy + 56.25 Gy)	59.25 Gy (28 Gy +31.25 Gy)
Akiba et al. [30]	150 Gy (75 Gy + 75 Gy)	78 Gy (39 Gy + 39 Gy)
Son et al. [31]	119.79 Gy (78.75 Gy + 41.04 Gy)	69.238 Gy (43.75 Gy + 25.488 Gy)
Ozgen et al. [32]	129.5 Gy (110–150 Gy)	-
Scharp et al. [33]	100 Gy (60 Gy + 40 Gy)	60 Gy (36 Gy + 24 Gy)
Minniti et al. [34]	131.25 Gy (75 Gy + 56.25 Gy)	70.25 Gy (39 Gy + 31.25 Gy)
Aktan et al. [35]	137.5 Gy (110.5–150 Gy)	-
Guo et al. [36]	115 Gy (75 Gy + 40 Gy)	63 Gy (39 Gy + 24 Gy)
Logie et al. [37]	120 Gy (60 Gy + 60 Gy)	56 Gy (28 Gy + 28 Gy)
Bernhardt et al. [38]	100 Gy (60 Gy + 40 Gy)	60 Gy (36 Gy + 24 Gy)
Suzuki et al. [39]	115 Gy (75 Gy + 40 Gy)	63 Gy (39 Gy + 24 Gy)

Abbreviations: BED, biological effective dose.

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
