# Peer review of "Re-Whole Brain Radiotherapy May Be One of the Treatment Choices for Symptomatic Brain Metastases Patients"

_cancers, 2022, doi:10.3390/cancers14215293_

Round 1
Reviewer 1 Report
Review report
Title: The review of re-whole brain radiotherapy
Authors:Takashi Ono and Kenji Nemoto
This paper reviews the re-whole brain radiotherapy administration among patients with brain metastases recurrence. The author summarizes a series of studies article and starts the article with the following aspects: treatment efficacy, overall survival and prognostic factors, toxicities,dose prescription and total dose,steroid administration,concurrent chemotherapy and meningeal dissemination.This article gives us a better understanding of the effect of re-whole brain radiotherapy for patients with brain metastases recurrence.
However, the following issue should be addressed:
(1) The title does not reflects the subject of the study. Please reformulate the title of the article.
(2) Please reduce the use of redundant words, e.g.however, although. I suggest inviting some native speakers to polish the manuscript.
(3) This article mainly reviewed the re-WBRT administration, but the author used "WBRT" at the end of introduction and conclusion part. I think this is inconsistent with the content of the article. Please check it.
(4) The author presents that “However, Hazuka et al.reported no complete resolution with short survival.” on line 104 and “ Suzuki et al. reported that patients receiving radiotherapy had better local control than those receiving the best supportive” on line 119. I believe that if the concrete data could be described in the review, the content of the article will be more perfect.
(5) The selection of re-WBRT treatment should consider the patient's previous treatment status and the patient's symptoms. The application time of re-WBRT will be greatly affected, and the treatment related factors are not discussed enough. Please discuss them in detail.
Author Response
October 20, 2022
To Prof. Dr. Samuel C. Mok, and reviewers,
First of all, we would like to thank all of you for the prompt review process and for giving us comments and suggestions about our manuscript. We really appreciate your kind cooperation and great advice to make our manuscript more sophisticated.
On behalf of all authors, I would like to re-submit the enclosed revised manuscript, entitled “The review of re-whole brain radiotherapy” for consideration for possible publication in Cancers.
We have responded to the reviewer’s comments as follows. Changes of manuscript highlighted in yellow in the revised manuscript.
We hope this revision process remedied all of concerns of the previous manuscript.
(1) The title does not reflects the subject of the study. Please reformulate the title of the article.
→ Thank you for the great comment. We rewrite the title as “Re-whole brain radiotherapy may be one of the treatment choices for symptomatic brain metastases patients”.
(2) Please reduce the use of redundant words, e.g.however, although. I suggest inviting some native speakers to polish the manuscript.
→ Thank you for the helpful comment. We asked Ediatage English proofreading again.
(3) This article mainly reviewed the re-WBRT administration, but the author used "WBRT" at the end of introduction and conclusion part. I think this is inconsistent with the content of the article. Please check it.
→ Thank you for the helpful comment. These were our mistake. We revised them (line 67 of page 2, line 361 of page 10).
(4) The author presents that “However, Hazuka et al.reported no complete resolution with short survival.” on line 104 and “ Suzuki et al. reported that patients receiving radiotherapy had better local control than those receiving the best supportive” on line 119. I believe that if the concrete data could be described in the review, the content of the article will be more perfect.
→ Thank you for the good comment. We added following sentences.
(only one patient survived more than 1 year with the median survival time of 8 weeks) (lines 111-112 of page 4)
(hazard ratio was 0.533, p = 0.050)
(line 124 of page 5)
(5) The selection of re-WBRT treatment should consider the patient's previous treatment status and the patient's symptoms. The application time of re-WBRT will be greatly affected, and the treatment related factors are not discussed enough. Please discuss them in detail.
→ Thank you for the helpful comment. We wrote about prognostic factor for re-WBRT in “3. Overall survival and prognostic factors” section including “time between the first WBRT and re-WBRT” (did you use application time as the time between first WBRT and re-WBRT?) (lines 146 – 190 of pages 5 - 6). As far as we know, there was no more data which we showed in this section. So, we cannot discuss more detail. Instead of it, we added concrete data as follows for readers to understand more easily. If you use “application time” as the preparing time for re-WBRT, there were no data as far as we know.
(< 60–80) (line 150 of page 5)
Sincerely,
Takashi Ono
Department of Radiation Oncology, Yamagata University Faculty of Medicine, 2-2-2, Iida-Nishi, Yamagata, Japan, 990-9585.
Tel.: +81-23-628-5386; Fax: +81-23-628-5389.
Email: abc1123513@gmail.com
Reviewer 2 Report
This paper has comprehensively reviewed re-WBRT.
It is well-written, although some English correction seems to be needed.
Line 133: I think "5 years" is misplaced. please rephrase the sentence.
In conclusions, not "WBRT may improve symptoms ..." should be corrected to "Re-WBRT" (In line 313, 339, 359 and 361)
Author Response
October 20, 2022
To Prof. Dr. Samuel C. Mok, and reviewers,
First of all, we would like to thank all of you for the prompt review process and for giving us comments and suggestions about our manuscript. We really appreciate your kind cooperation and great advice to make our manuscript more sophisticated.
On behalf of all authors, I would like to re-submit the enclosed revised manuscript, entitled “The review of re-whole brain radiotherapy” for consideration for possible publication in Cancers.
We have responded to the reviewer’s comments as follows. Changes of manuscript highlighted in yellow in the revised manuscript.
We hope this revision process remedied all of concerns of the previous manuscript.
It is well-written, although some English correction seems to be needed.
Line 133: I think "5 years" is misplaced. please rephrase the sentence.
In conclusions, not "WBRT may improve symptoms ..." should be corrected to "Re-WBRT" (In line 313, 339, 359 and 361)
→ Thank you for the helpful comment. This was our mistake. We removed or replaced them (line 137 of page 5. line 318 of page 9. line 344, 364, and 366 of page 10).
Sincerely,
Takashi Ono
Department of Radiation Oncology, Yamagata University Faculty of Medicine, 2-2-2, Iida-Nishi, Yamagata, Japan, 990-9585.
Tel.: +81-23-628-5386; Fax: +81-23-628-5389.
Email: abc1123513@gmail.com
Round 2
Reviewer 1 Report
I recommend accepting in present form.